# Wind Speed and Landscape Context Mediate *Campylobacter* Risk among Poultry Reared in Open Environments

**DOI:** 10.3390/ani13030492

**Published:** 2023-01-31

**Authors:** Olivia M. Smith, Kevin A. Cornell, Michael S. Crossley, Rocio Crespo, Matthew S. Jones, William E. Snyder, Jeb P. Owen

**Affiliations:** 1School of Biological Sciences, Washington State University, Pullman, WA 99164, USA; 2Department of Entomology, University of Georgia, Athens, GA 30602, USA; 3Ecology, Evolution, and Behavior Program, Michigan State University, East Lansing, MI 48824, USA; 4Avian Health and Food Safety Laboratory, Washington State University, Puyallup, WA 98371, USA; 5Department of Population Health and Pathobiology, College of Veterinary Medicine, North Carolina State University, Raleigh, NC 27606, USA; 6Department of Entomology, Washington State University, Pullman, WA 99164, USA; 7Cascade Agroecology, Leavenworth, WA 98826, USA

**Keywords:** campylobacter, chickens, extensified farming, free range, foodborne illness, open environment, organic farming, poultry disease

## Abstract

**Simple Summary:**

*Campylobacter* spp. from contaminated chickens is the single largest cause of foodborne illness in the United States. Returning to traditional farming practices, such as rearing poultry on rotational pastures, is proposed to reduce human disease risks while improving animal welfare. Yet, little is understood about how environmental factors and farm management alter pathogen prevalence. Here, we assess environmental and farm management factors that impact *Campylobacter* spp. prevalence in poultry on 27 open-environment farms spanning the USA states of California, Oregon, Washington, and Idaho over a three-year period. We found that 26.0% of individual chicken fecal samples we tested were positive for *Campylobacter* spp. We detected *Campylobacter* spp. in ≥1 fecal sample in 69.4% of flocks and 85.2% of farms. Our analyses suggest that *Campylobacter* spp. prevalence increases in poultry on farms with higher average wind speeds in the seven days preceding sampling; on farms embedded in more agricultural landscapes; and in flocks typified by younger birds, more rotations, higher flock densities, and the production of broilers. Our findings indicate that farmers will face greater risk of infected poultry when there are high wind speeds, when their farms are located in highly agricultural areas, and if they raise broilers.

**Abstract:**

Foodborne pathogens cause over 9 million illnesses in the United States each year, and *Campylobacter* from chickens is the largest contributor. Rearing poultry outdoors without the use of antibiotics is becoming an increasingly popular style of farming; however, little is understood about how environmental factors and farm management alter pathogen prevalence. Our survey of 27 farms in California, Oregon, Washington, and Idaho, USA, revealed a diversity of management practices used to rear poultry in the open environment. Here, we assess environmental and management factors that impact *Campylobacter* spp. prevalence in 962 individual chicken fecal samples from 62 flocks over a three-year period. We detected *Campylobacter* spp. in 250/962 (26.0%) of fecal samples screened, in 69.4% (43/62) of flocks, and on 85.2% (23/27) of farms. We found that *Campylobacter* spp. prevalence was predicted to increase in poultry on farms with higher average wind speeds in the seven days preceding sampling; on farms embedded in more agricultural landscapes; and in flocks typified by younger birds, more rotations, higher flock densities, and the production of broilers. Collectively, our results suggest that farms in areas with higher wind speeds and more surrounding agriculture face greater risk of *Campylobacter* spp. introduction into their flocks.

## 1. Introduction

*Campylobacter* in poultry is the largest single contributor to foodborne illness in the United States [1]. Modern commercial poultry production—including both egg-laying (“layers”) and meat (“broilers”) birds—use large indoor facilities to rear high-density flocks that are susceptible to disease outbreaks [2]. With the intention to protect flocks from disease, conventional growers have adopted strict biosecurity practices: the routine sterilizing of chicken housing, antibiotic treatments, and separating flocks from the surrounding environment [3,4,5]. However, these practices have recently engendered public concerns for animal health and welfare [6,7,8,9]. Increased consumer preference for organic, local, and certified humane animal products are driving greater demand for poultry products from “open-environment” rearing practices, including free-range, pastured, and organic production [10,11,12]. These practices represent a variety of management styles that allow flocks to have access to the outdoors [13]. While there are many perceived benefits to integrating livestock into outdoor environments, poultry are potentially exposed to pathogens in the environment (e.g., soil) or via contact with wildlife. These routes of pathogen exposure may affect the prevalence of infectious disease and impact grower production and human health.

*Campylobacter* is highly prevalent in poultry around the world [14,15,16]. It is a commensal bacterium that spreads through fecal–oral contamination and can persist in the bird’s gastrointestinal tract without causing symptoms [17]. Rearing poultry outdoors potentially increases the risk of introducing pathogens into the flock from water, soil, other livestock, or wildlife [18]. Prior research has demonstrated that foodborne pathogens are more likely to be detected on farms near grazeable land and where cropland expands at the expense of seminatural vegetation [19]. Therefore, poultry produced in landscapes with greater extents of agriculture may have greater risk of exposure to *Campylobacter* (Figure 1). Agricultural intensification may increase exposure risk by increasing pathogen abundance in water, prevalence in livestock, or through mechanisms such as increasing pathogen prevalence in wild birds that then contact poultry [19,20,21]. Wild birds, also reservoirs of *Campylobacter* spp. [22], can reach high densities around livestock on extensified farms [21,23] and could introduce *Campylobacter* to flocks [24]. Environmental factors such as high wind speeds can further impact disease risk by introducing greater amounts of pathogens found in surrounding landscapes [18]. 

Once *Campylobacter* is introduced into a flock, the dynamics of the flock may affect its epizootiology. Open-environment rearing practices include a variety of management decisions that influence these dynamics, such as variation in flock size, poultry age, poultry breed(s), density, flock movement (rotation), and pasture space. These factors are potential mediators of *Campylobacter* occurrence and transmission. The transmission of *Campylobacter* can be magnified in larger flocks and at higher flock densities [25,26]. Spatial management of flocks found in open-environment rearing may alter pathogen transmission [13,27]. One popular management practice is rotational rearing, where growers move flocks to fresh pasture at different intervals to maximize feed available from herbaceous vegetation, improve land fertility by distributing nutrients from chicken feces, and protect farmland from desiccation [28,29,30]. These rotations may also be incidentally reducing exposure of poultry to *Campylobacter* by moving flocks away from contaminated ground, which has been shown in other grazing systems to avoid fecal–oral parasites [31]. However, small farm areas may negate any disease-reducing effects of rotating flocks if contaminated ground is unavoidable. In addition, soil properties, such as organic matter, can impact the survival of bacteria and complicate the effects of flock management on pathogen exposure [32]. Various combinations of these factors may influence the prevalence of *Campylobacter* in flocks reared in open-environment systems, but this has not been fully studied among diverse farm types.

Here, we assess factors that impact *Campylobacter* spp. prevalence in 962 chicken fecal samples from 27 open-environment farms spanning California, Oregon, Washington, and Idaho in the USA over a 3-year period (Figure 2). Our objectives were to survey for prevalence of *Campylobacter* spp. and to examine the relative influence of landscape, weather, wild bird–poultry interactions, and farm management on *Campylobacter* spp. prevalence in chickens reared in open environments (Figure 1). To address our objectives, we performed bacterial assays on chicken feces; conducted wild bird point count surveys; collected farm management data and classified farm management using principal component analysis; collected local weather station data; and classified landscape context using remotely sensed data.

## 2. Materials and Methods

### 2.1. Farm Surveys

We surveyed 27 open-environment poultry farms in California (*n* = 7), Oregon (*n* = 2), Washington (*n* = 17), and Idaho (*n* = 1), USA (Figure 2 and Appendix A). One farm managed 2 locations and is counted as one farm in summaries, but our analyses use each location’s unique management and landscape variables. We conducted one survey per farm per year between May and August 2017–2019. Of the 27 farms, one farm was visited for only one summer, 18 farms were visited for two, and eight farms for three (Figure 2). The year-to-year variation was due to variation in grower recruitment and continuing participation. Some farms had multiple flocks, and some flocks were retained by farms between years. We selected farms based on the following criteria: (1) farms raised poultry outdoors; (2) farms did not use hormones, antibiotics, and prophylactic drugs; (3) flock space was accessible to wild birds; and (4) farmers granted us access to survey their flocks. 

### 2.2. Bacterial Testing 

On each farm, we collected chicken feces from the environment that we sent to the Washington Animal Disease and Diagnostics Laboratory (WADDL; 2017 and 2019) or the Avian Health and Food Safety Laboratory (AHFSL; 2017 and 2018) to culture for *Campylobacter* spp. (2017–2019) and *Salmonella* spp. (2017–2018). Each sample was only tested once by one laboratory, but we had to alter laboratories used between surveys due to laboratory capacity and transportation. We only collected feces that were visibly isolated (i.e., not in contact with other fecal samples) and not “trampled” to minimize the likelihood that samples came from the same bird or were cross contaminated, respectively. We collected visibly fresh fecal samples to minimize variation in sample age or changes in the bacterial community over time. During collections, we haphazardly walked a zig-zag pattern within each flock’s pen or foraging area. For each sample, we scraped off and collected the top ~2.5 cm of each fecal sample to avoid contamination from the ground. The number of samples collected per survey ranged from a minimum of 5 for flocks of 10 birds, to a maximum of 30 for flocks of 300 birds or greater. Fecal samples were immediately chilled at 4 °C on wet ice and shipped overnight to the Washington State University diagnostic laboratories for bacterial screening and quantification. Samples were screened no later than 48 h after collection from the farm. 

We tested feces for *Campylobacter* spp. using microaerophilic cultures, latex agglutination, and biochemical confirmation. First, we incubated samples at 37 °C for 24 h under microaerophilic conditions. After incubation, we plated samples on *Campylobacter* CVA agar (Cefoperazone, Vancomycin, and Amphotericin B Agar with 5% Sheep Blood, Hardy Diagnostics, Santa Maria, CA, USA) plates and incubated for another 24 h. Plates were then removed from the incubator to be examined for growth. If the culture showed no new growth, it was re-cultured and monitored over a 72 h period. The suspect *Campylobacter* isolates were then examined for morphological characteristics (e.g., S-shaped, seagull shaped, long spiral forms, and darting motility). If we identified suspect colonies, we performed a latex agglutination (catalog #F46, Hardy Diagnostics, Santa Maria, CA, USA) test for confirmation. Additionally, for samples collected in 2017 and 2019 that were sent to WADDL, we used an API *Campylobacter* biochemical test kit (BioMérieux, API Reference Guide, Hazzlewood, MO, USA) to identify *Campylobacter* isolates at the species level. 

Aerobic culturing of *Salmonella* spp. was conducted in 2017 and 2018 to detect and identify the presence of *Salmonella* spp. in each fecal sample. We discontinued sampling in 2019 due to the low detection rate of *Salmonella* spp. and high cost of culturing. Culturing methods employed followed Dufour-Zavala [33]. Briefly, a subsample of the homogenized fecal sample was incubated at 37 °C for 18 to 24 h followed by an aerobic culture. If *Salmonella* spp. were detected, serological identifications were made using slide agglutination by somatic antigen (catalog# 90001-672, VWR International Inc., Radnor, PA, USA) and the poly H (Difco 224061, catalog# 90001-760, VWR International Inc., Radnor, PA, USA) procedure. 

### 2.3. Landscape Characteristics

We used the National Land Cover Database (NLCD) to calculate percent natural/semi-natural land cover (“natural”; includes forest [values 41–43], scrubland [values 51–52], herbaceous [values 71–74], and wetland categories [values 90–95]) and percent agricultural land cover (pasture/hay [value 81] and cultivated crops [value 82]) in a 2.1 km buffer. The percent natural and percent agricultural land cover were strongly inversely correlated (Pearson’s correlation = −0.8, *p* < 0.0001), so we only used percent agricultural land cover in our analyses. The NLCD database has a 30 m resolution with estimated 82–88% overall accuracy [34]. We selected these classifications and a 2.1 km buffer following Smith et al. [21], wherein the authors found that *Campylobacter* spp. prevalence in wild bird feces from production areas increased in relation to agricultural intensification and decreased with increasing amounts of natural cover in a 2.1 km radius. We followed Smith et al. [21] because we hypothesized that landscape context may primarily mediate *Campylobacter* spp. prevalence in poultry via mediating wild bird community competence of the pathogen. However, the landscape context may also mediate *Campylobacter* spp. prevalence in poultry if pathogens are blown in from other farms by wind or are brought in through runoff [35]. 

### 2.4. Wild Bird Surveys

We conducted bird contact surveys using a modified point count protocol [23] to estimate contact rates with chickens. The bird contact counts were conducted by two trained ornithologists proficient at visual and auditory identification of all bird species common to the study region. We conducted all counts at chicken coops or at nearby vantage points where the surveyor could clearly see the entire poultry enclosure without disturbing wild bird activity. Each survey was conducted between sunrise and 10 am for a duration of 10 min. Surveys were performed within one day of chicken fecal sample collection. Only birds that entered the poultry enclosure or perched on enclosure fencing were included in our analyses. If there was no physical enclosure at the poultry coop, we only counted birds that entered the “foraging space” (see “2.7, Flock Management” below). We recorded the number of unique individuals per species observed using the coop area, including aerial foragers [23]. We excluded birds that flew over the coop following standard point count protocols. 

The survey area was equal to the poultry enclosure size (see “2.7, Flock Management” below), which varied between farms. Therefore, we divided total wild bird contacts over the 10 min survey period by coop enclosure size for a per-area contact metric. In 12/63 of our point counts, flocks were adjacent (one adjacent count covered three flocks, while the other adjacent counts covered two flocks). Therefore, we conducted one count and divided contacts by the combined enclosure area. Additionally, three broiler flocks were housed in covered pens but were rotated on pasture where they could be exposed to wild bird feces (e.g., Appendix A). Therefore, they received a contact score of 0, but we accounted for possible exposure during rotations through a flock rotation intercept (farm rotates flocks 1+ times per year or does not) in all models that included wild bird contact (see 2.8, Statistical Analyses). 

### 2.5. Weather Variables

Weather variables were selected by referencing Golden et al. [36] and Hwang et al. [18], who identified important weather predictors influencing enteric pathogen prevalence in pastured poultry. For each farm, we found the nearest weather station on Weather Underground (https://www.wunderground.com/wundermap, accessed on 29 December 2020). We then obtained daily mean temperature, humidity (%), and wind speed from the closest weather station on Weather Underground for the seven days preceding fecal sample collection. We then averaged the values across the seven days for our analyses. 

### 2.6. Soil Sampling and Analysis

Physical soil samples were collected on farms at locations where poultry resided. Soil samples were collected by extracting 8” diameter cores of the top 6” of topsoil. We collected three core samples per flock on pasture actively used by chickens and then homogenized samples for physical testing. Each soil sample was analyzed by SoilTest Farm Consultants (Moses Lake, WA, USA) for physical and chemical properties using standard methods. We calculated organic matter using standard weight “loss on ignition” calculated as a percentage of the total soil content [37]. We were unable to collect soil samples from three farms. 

### 2.7. Flock Management

We collected data on flock management (e.g., total pasture size on the farm, enclosure size, flock size, rotations per year, average poultry age) using a combination of visual inspections and a farmer questionnaire survey, previously described in [13]. We estimated total farm pasture size in ArcGIS by identifying the entire area throughout the year where each farm kept flocks (version 10.7; ESRI, Redlands, CA, USA). We also estimated each flock’s enclosure size at the time of the survey, which was defined as the amount of space within the fencing that contained the poultry, also using ArcGIS. The difference between pasture size and enclosure size was that pasture size included space where poultry could be rotated into, whereas enclosure size represented the space fenced off for poultry during a given survey period. Next, flock density was calculated as the number of individuals within a flock (flock size) divided by the flock’s available enclosed space at the time of the survey (enclosure size). We estimated rotations per year by asking farmers how often they moved flocks throughout the year (e.g., once a week or never). We also estimated the average flock age by consulting with farmers. Most flocks (*n* = 44) contained a cohort of single aged birds, but some flocks (*n* = 18) consisted of mixed ages. We estimated the average age of mixed-cohort flocks by estimating the percentage of each flock containing each age cohort of birds then weighting each cohort’s age reported to us by the farmers by their representation within the flock. 

### 2.8. Statistical Analyses

Because farm management variables were correlated (Appendix A), we used principle component analysis (PCA) to assess relationships among farm management variables using the prcomp function in R version 3.6.3 [38], as previously described in [13]. To account for the variation in scale among metrics in the PCA, variables were z-score transformed prior to analyses to improve model convergence by taking [(X−$\bar{X}$)/sd], where X is the raw data for the sample, $\bar{X}$ is the mean value calculated across all samples, and sd is the standard deviation for the variable calculated across all samples. The first four principal components (PC) combined accounted for 91.3% of the variation: PC1, PC2, PC3, and PC4 accounted for 36.7%, 26.8%, 16.0%, and 11.8% of the variation, respectively. We used the first two PCs in subsequent analyses. Increasing values of PC1 were associated with younger birds, more rotations per year, and higher flock densities, which we label as “production type PC1” (Figure 3). Flocks with low PC1 values were typically egg-laying hens, while high values of PC1 were typically broiler flocks. Increasing values of PC2 were primarily associated with larger enclosure sizes, larger pastures, and larger flocks, which we label as “production space PC2.” 

To determine the relative importance of landscape context (% agriculture), weather (average temperature, wind speed, and humidity in the seven days prior to sampling), wild bird contact, and farm management (production type PC1, production space PC2) on *Campylobacter* spp. prevalence, we used generalized linear mixed models via the glmmTMB package in R [39] and AIC_c_ model selection in the bblme package in R [40]. Our candidate model set included potential single, additive, and interactive effects of on- and off-farm variables hypothesized to influence *Campylobacter* spp. prevalence in poultry feces, in addition to a null model (see Appendix A for all candidate models). Our models used a betabinomial distribution and logit link function, with the number of fecal samples that were positive and negative as the response variable. We used the betabinomial distribution to account for overdispersion in the data and because it was a better fit than the binominal (lower AIC_c_). We used flock nested within farm as a random effect to account for multiple visits to the same flocks and farms. All continuous variables were standardized prior to analyses by subtracting each value from the mean and dividing by the standard deviation of that variable (i.e., variables were z-score transformed), except for the two PCs that were already scaled. We assessed multicollinearity for all candidate models in the performance package in R and removed models with variance inflation factors above five from further consideration [41]. We ranked models based on AIC_c_ and identified those that were most well-supported based on a criteria of ∆AIC_c_ < 2.0 [42]. 

Because our model selection approach requires complete data (no missing values), we conducted a separate set of analyses to examine if soil organic matter (which had missing observations) should be considered in full analyses by re-running analyses on the subset of farms with soil parameters (Appendix A). Models including soil organic matter did not have high support (first model with soil organic matter ∆AIC_c_ = 6.9), so soil was excluded from our final analyses. 

## 3. Results

### 3.1. Farm Characteristics

Nineteen of twenty-seven (70.4%) farms raised other livestock (e.g., pigs, goats, sheep, cattle, horses, donkeys) in addition to poultry, 16/27 (59.2%) grew crops in addition to livestock, and 11/27 (40.7%) were USDA certified organic producers. The total farm size averaged 25.6 ha ± 9.5 (SE). The frequency of biosecurity practices on farms in our study can be found in Appendix A. Some of the most commonly used practices included fencing, the cleaning of housing, and vermin-proof bins. Organic matter ranged from 2.6 to 12.1% in soil (mean: 7.0% ± 0.3 (SE)), and available water capacity ranged from 5 to 23 (mean: 16.1 ± 0.7 (SE)). The average age of broiler flocks was 11.0 wk (±1.1 SE), and the average age of layer flocks was 84.9 (±7.67 SE) wk. Among 121 species of wild birds observed on the farms in this study, 42 were recorded interacting with poultry flocks. The most common wild bird species were the barn swallow (*Hirundo rustica*), European starling (*Sturnus vulgaris*), and house sparrow (*Passer domesticus)*.

### 3.2. Bacteria Detection 

We detected *Campylobacter* spp. in 250/962 (26.0%) fecal samples screened (Table 1), in 69.4% (43/62) of flocks, and on 85.2% (23/27) of farms. *Campylobacter* spp.-positive farms were located in all states surveyed (Appendix A). For the two years that we identified *Campylobacter* at the species level, we detected *C. jejuni* in 24/126 (19.0%; 2017) and 97/480 (20.2%, 2019) fecal samples. *C. coli* was less common and was detected in 0/126 (0%; 2017) and 10/480 (2.1%, 2019) fecal samples. For the two years we screened for *Salmonella* spp., we detected an overall prevalence of 1.7% (8/482 fecal samples), and all positive samples were collected in 2018 from just three farms. Since *Salmonella* spp. were so rarely detected, we did not examine correlations with environmental factors nor *Campylobacter* spp.

### 3.3. Mediators of Campylobacter spp. Prevalence

*Campylobacter* spp. prevalence was predicted to increase on farms with higher average wind speeds in the seven days preceding sampling (β = 0.52, 95% CI: 0.21, 0.83; Figure 4a and Appendix A). For every 1 kph increase in wind speed, the odds of detecting *Campylobacter* spp. increased by 1.2%. Additionally, poultry flocks with higher production type PC1 values (typically younger birds, more rotations per year, higher flock densities, and broiler flocks) were predicted to have higher *Campylobacter* spp. prevalence (β = 0.29, 95% CI: 0.07, 0.51; Figure 4b). Finally, flocks on farms embedded in landscapes with higher amounts of agricultural land cover were predicted to have higher *Campylobacter* spp. prevalence (β = 0.52, 95% CI: 0.15, 0.90; Figure 4c). For every 1.0% increase in % agricultural land cover, the odds of detecting *Campylobacter* spp. increased by 1.0%. No other variables (i.e., wild bird contact, temperature, humidity, production space PC2, rotation; Appendix A) appeared in models with high support (i.e., ∆AIC_c_ < 2.0).

## 4. Discussion

The management of poultry raised in open environments creates complex vulnerabilities to environmental pathogens [6,43]. *Campylobacter* from poultry is a common cause of human illness [1]; thus, understanding how to reduce poultry farm contamination can reduce human disease risk. Our study across diverse open-environment poultry farms detected *Campylobacter* spp. in 26.0% of all individual fecal samples collected. In the years that we identified *Campylobacter* at the species level, *C. jejuni* was the most common (in 20.0% of samples), followed by *C. coli* (in 1.7% of samples). We detected *Campylobacter* spp. on 85.2% (23/27) of farms and in 69.4% (43/62) of flocks, suggesting a high prevalence among farms and their environments but a relatively lower prevalence within individuals of those flocks. Estimated prevalence in this study is similar to “backyard” and rural flocks from other regions globally, such as 18.3% in Italy [44] and 31.9% in Vietnam [45]. We did not survey large hen houses that used conventional growing methods for this study; however, a meta-analysis performed by Rossler et al. [16] found a higher prevalence of *C. jejuni* in hens (39.5%) and broilers (32.2%) than we did. Rossler et al. [16] also found *C. coli* to be rarer than *C. jejuni* (between 10–17%). To our knowledge, the data reported here represent the broadest geographic survey for *Campylobacter* spp. prevalence among open-environment poultry farms conducted to date. 

We also examined if weather, landscape context, farm management, soil organic matter, or wild bird contact predicted the prevalence of *Campylobacter* spp. in open-environment poultry flocks. First, average wind speed collected from local weather stations was a positive predictor of *Campylobacter* spp. detection in individual poultry samples. This is consistent with the recent findings of Hwang et al. [18], who showed that wind speed was a positive predictor of *Salmonella* spp. prevalence, and Golden et al. [36], who found that wind was positively correlated with *Listeria* spp. colonies detected in feces and soil samples in pastured systems. Wind may introduce contaminated material (e.g., soil and dried fecal particles) onto farms, exposing flocks to *Campylobacter* spp., among other fecal pathogens, if flocks have access to the open environment. We also found that % agricultural land cover in the landscape had a positive relationship with *Campylobacter* spp. prevalence. Landscapes with a greater agricultural extent have been shown to have higher environmental contamination of enteric pathogens [19,21]. Intuitively, a combined effect of greater environmental contamination facilitated by faster average wind speeds will circulate *Campylobacter* into neighboring areas, as indicated by our models. 

We did not find a strong association between wild bird contact and *Campylobacter* spp. prevalence in poultry flocks. Wild birds are suspected sources of *Campylobacter* and *Salmonella* infections [6,46], and there is evidence that wild birds actively contact poultry on open-environment farms [47], as we observed in this study. Although feces collected from wild birds have higher *Campylobacter* spp. prevalence in more agricultural areas, the potential for wild bird introduction of bacteria into a poultry flock is affected by both the wild bird abundance and the species’ traits (e.g., reservoir competence and behavior) [21,48]. We did not model *Campylobacter* spp. risk by contacts with wild bird species with different traits or individual wild bird species because it was outside the scope of this study. Rather, we were interested in testing if the total abundance of wild birds interacting with poultry had any effect on *Campylobacter* spp. prevalence. We suspect that our quantification of total contact/enclosure area may mask important wild bird identity or trait effects that influence the transmission of *Campylobacter* spp. [48]. Thus, we cannot rule out wild birds as an important mediator of *Campylobacter* spp. on open-environment farms, and the two-way transmission of *Campylobacter* spp. between outdoor poultry and wild birds may impact wild bird conservation and food safety [46]. It is also of note that although we did not find a relationship between wild bird contact and *Campylobacter* spp. prevalence, we did find a relationship between landscape context and *Campylobacter* spp. prevalence. It is possible that the landscape effect emerged as a result of shifts in wild bird communities towards more competent hosts in more agricultural landscapes, and thus, the landscape context could be a better proxy for wild bird community risk in pathogen transmission than wild bird contact alone [21]. Future work should focus on characterizing transmission risk posed by wild birds as a function of bird traits (reservoir competence), poultry contact (behavior), and landscape context.

We observed that flocks typified by younger birds, more rotations per year, higher flock densities, and broiler breeds had higher *Campylobacter* spp. prevalence. These flock management variables grouped together (production type PC1), which makes it difficult to disentangle the possible driver(s) of *Campylobacter* spp. prevalence. Several factors in production type PC1 intuitively associate with pathogen transmission: higher flock densities are expected to increase pathogen transmission [49]; younger birds may have increased susceptibility to infection and reduced immune function to clear *Campylobacter* spp. infections, which has been suggested to improve as chickens age [50]; increased flock rotations could reduce pathogen exposure by increasing the spatial separation of chickens from sources of bacteria [43]; and *Campylobacter* spp. has been described to rapidly colonize to nearly 100% prevalence in broiler flocks near the age of slaughter [50]. It may be that the intensifying effects of younger birds and higher flock density, common in broiler production, overwhelm the expected dampening effects of flock rotation on *Campylobacter* spp. prevalence. The production space PC2 (larger enclosure sizes, larger pastures, and larger flocks) was not associated with *Campylobacter* spp. prevalence, which suggests that farm area and numbers of chickens are not more important than flock density and chicken age. Given that free-ranging poultry naturally congregate around water, supplemental feed, and refugia, it may be that increasing the amount of foraging area has little effect on the actual densities of chickens in an open environment [13]. Overall, these results underscore the complex combinations of management variables that are represented across farms using open-environment production.

Our study faced two important limitations that may affect interpretations of the results. First, our sampling of each farm occurred at a single time-point per year. This was necessary because of the wide geographic area that was included in the study and the limitations in personnel and sampling costs. By constraining analyses to a single survey/farm/year, we may have missed seasonal fluctuations in *Campylobacter* spp. prevalence, management, and wild bird activity. Importantly, despite this sampling constraint, we were able to visit 24/27 farms over two or more consecutive years, and 17/62 flocks were retained and sampled again the following year. Thus, there was continuity in both sampling environment and flocks across the three-year study. Second, our study is correlational and does not allow for the clear separation of causative factors associated with *Campylobacter* spp. prevalence. As discussed, there are multiple variables that appear to be associated with *Campylobacter* spp. prevalence, including wind, agricultural extent in the landscape, and management factors. This does not allow for clear guidance on how to limit *Campylobacter* spp. infections of poultry in open environments. However, it draws attention to the complexities of open-environment poultry farming and the diverse combinations of management approaches and environmental conditions that can affect disease risk. There may be no “simple solutions” to managing infectious diseases of poultry in open environments. Instead, the management of disease risk may require multiple approaches and case-by-case risk assessment based on weather patterns and the landscape context of a farm.

## 5. Conclusions

Our study of diverse open-environment poultry farms across a broad geographic range suggests that several important variables are associated with the prevalence of *Campylobacter* spp. in chicken flocks. There may be higher risk of *Campylobacter* spp. infections on farms located in areas with an increased agricultural extent. Farmers may benefit from the awareness and mitigation of wind (e.g., windbreaks) and should pay attention to flock health following high-wind-speed periods. Finally, farms with younger flocks and high flock densities may experience a higher prevalence of *Campylobacter* spp. infections. Future work should examine how neighboring livestock and presence of non-poultry livestock on farms influences prevalence of *C. jejuni* and *C. coli* in open-environment poultry, which was beyond the scope of our study.

## Figures and Tables

**Figure 1 animals-13-00492-f001:**
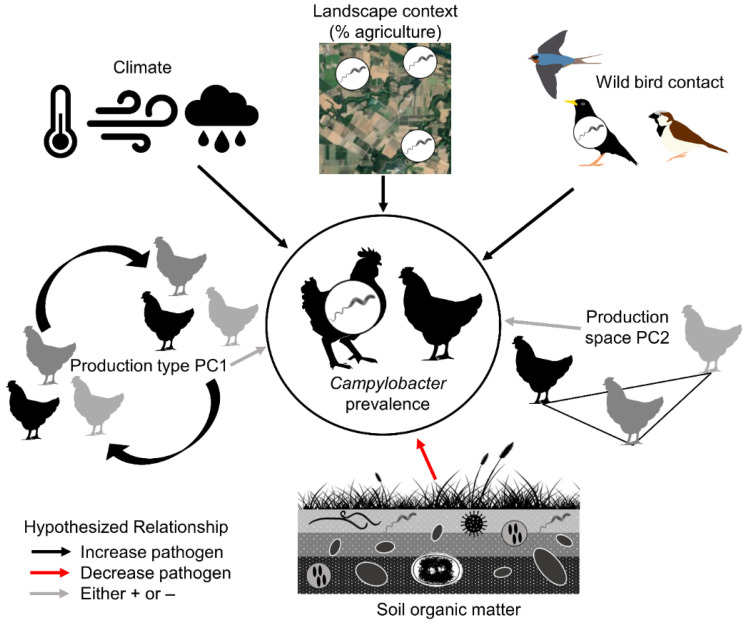
Hypothesized relationships between *Campylobacter* spp. prevalence, climate (temperature, wind speed, humidity), landscape context (% agriculture), wild bird contact, production type PC1, production space PC2, and soil organic matter. Arrows indicate if the hypothesized relationship is positive (black), negative (red), or uncertain (gray).

**Figure 2 animals-13-00492-f002:**
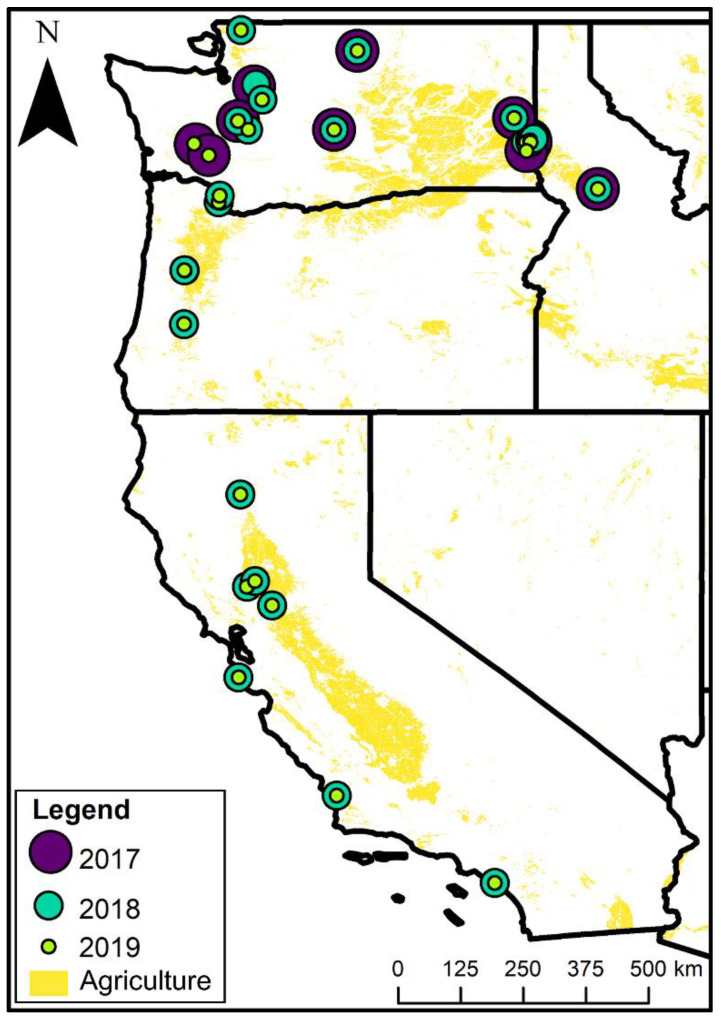
Map showing locations of farms surveyed each year of the study (2017–2019) in the USA states of California, Oregon, Washington, and Idaho. Year(s) each farm was surveyed indicated by color and size: large, purple circle = 2017; medium, teal circle = 2018; small, lime green circle = 2019. Yellow background indicates agricultural land use.

**Figure 3 animals-13-00492-f003:**
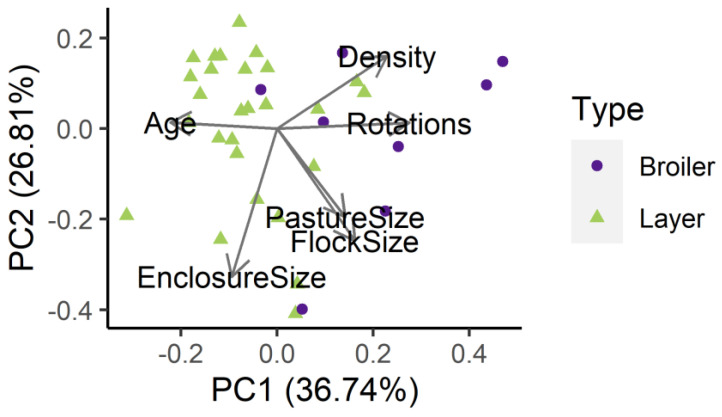
Principal components (PC) 1 and 2, showing variation in six farm management variables including pasture size (m^2^; space ever available to chickens throughout year), enclosure size (m^2^; area used by chickens at time of survey), flock size (number of chickens per flock), density (flock size/enclosure size), rotations per year (times moved per year), and average chicken age for the flock (months). Variation explained by each axis given in axis labels.

**Figure 4 animals-13-00492-f004:**
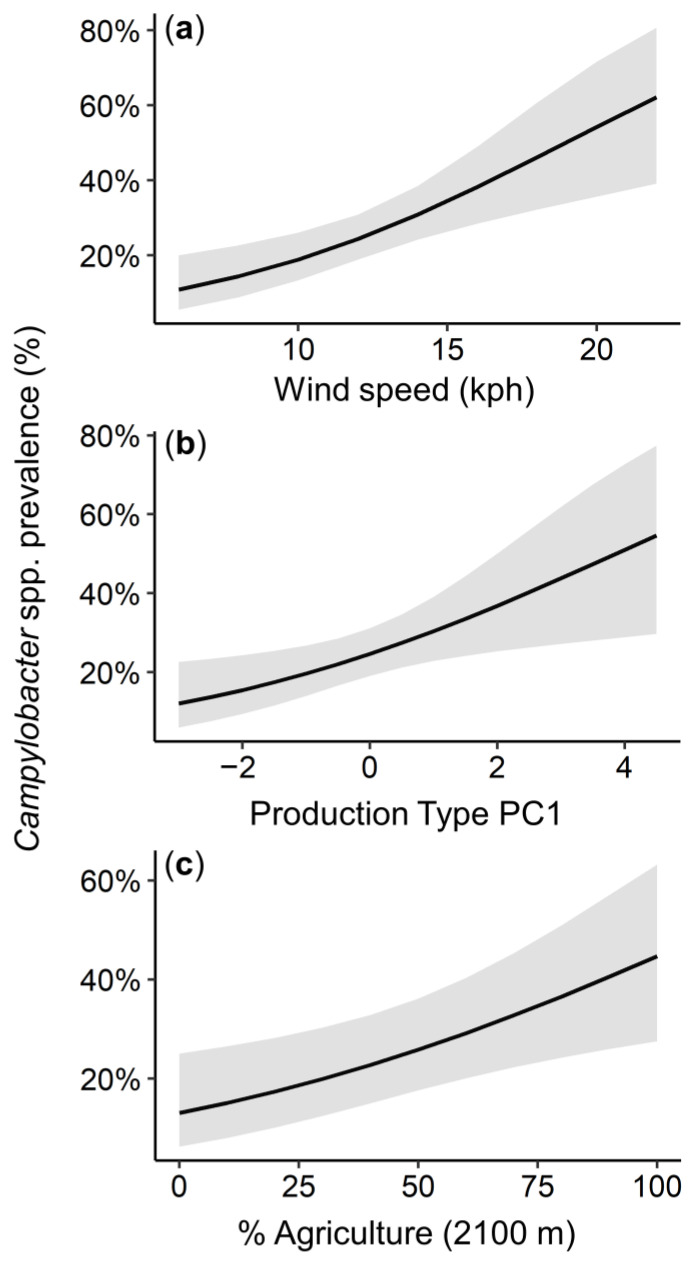
*Campylobacter* spp. prevalence was predicted to increase with (**a**) increasing wind speed (kph), (**b**) when flocks had high production type PC1 values, and (**c**) when farms were embedded in more agricultural landscapes (2100 m radius). Gray area shows +/− 95% confidence intervals.

**Table 1 animals-13-00492-t001:** Prevalence (number positive/number tested) of *Campylobacter* spp. and *Salmonella* spp. in 962 chicken fecal samples collected between 2017 and 2019 from poultry reared on 27 open-environment farms in California, Oregon, Washington, and Idaho, USA. Fecal samples were identified as *C. jejuni, C. coli,* or unknown in 2017 and 2019. Fecal samples were tested for *Salmonella* spp. from 2017 to 2018. N/A = not available.

Bacteria	Years	Positive/Tested (Prevalence)	*C. jejuni*	*C. coli*	Unknown *Campylobacter* spp.
*Campylobacter* spp.	2017–2019	250/962 (26.0%)	121/606 (20.0%)	10/606 (1.7%)	119/962 (12.4%)
	2017	54/126 (42.9%)	24/126 (19.0%)	0/126 (0%)	30/126 (23.8%)
	2018	80/356 (22.5%)	N/A	N/A	80/356 (22.5%)
	2019	116/480 (24.2%)	97/480 (20.2%)	10/480 (2.1%)	9/480 (1.9%)
*Salmonella* spp.	2017–2018	8/482 (1.7%)	N/A	N/A	N/A
	2017	0/126 (0%)	N/A	N/A	N/A
	2018	8/359 (2.2%)	N/A	N/A	N/A

## Data Availability

The data presented in this study are available in the article and its Appendix A.

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
