# Peer review of "Wind Speed and Landscape Context Mediate *Campylobacter* Risk among Poultry Reared in Open Environments"

_animals, 2023, doi:10.3390/ani13030492_

Round 1

Reviewer 1 Report

Interesting study on the risk associated with open farming systems, investigating the influence of surroundings (agriculture), wild birds, and weather on the spread and likelihood of Campylobacter infection. The study is well-written and commendable.

Methods

Ln 154: Suggest that ‘labs’ is replaced with ‘laboratories’ in keeping with the scientific format.

Ln 161, 166, 176, 177: Please provide the correct format for supplier information, thus city and country

Ln 164: Are “physiological” characteristics the correct terminology? Or is it rather the “morphological” characteristics?

Ln 168: Please insert the ‘bioMérieux’ supplier details for the API kits and complete as per previous comments.

Ln 194-196: Please clarify the statement: “Because… analyses”

Ln 204-205: The survey between sunrise and 1000hr, and the 10-minute period within 1 day collection is not clear. Please rephrase for clarity.

Suggest a number below ten to be written out. Please review the manuscript throughout. These appear very inconsistently throughout the manuscript, e.g. ln 205, 280, 296, 431.

It is suggested that the author/s use either none, one or two decimals consistently when reporting figures (%). These appear inconsistently throughout the manuscript, e.g. ln 271, 272, 336, 354, 361

The reviewer would like to raise a concern with the explanation of the methodology for the landscape statistical models (ln 279-297). The concern is selecting a model that may contribute to a hypothesis and thus be biased to a particular outcome, but looking at the supplementary data that may assist in that regard. However, the table needs more clarity.

At what age was the poultry when samples were collected? Often related to contamination levels.

Results

There are very little data related to actual p-values in the text that can support the correlation between agriculture and increased contamination levels of Campylobacter; also the correlation between increased stock levels or otherwise, although Figure 4 describes some confidence levels. The reviewer suggests that this is clarified.

Ln 359-360: This appears to be a grammatical error.

Discussion

A few thoughts that the reviewer would like to leave with the author/s: It is known that C. jejuni is the most prevalent food-borne bacteria when causing campylobacteriosis in humans. Following this, C. jejuni is also the most prevalent in poultry which makes the former statement more likely. What the authors have shown is that even in the environment, the contamination of poultry with C. coli is unlikely. What would have been very interesting is the sort of agriculture surrounding these farms. If other animals are present in neighbouring farms, either cattle (possible vector for C. jejuni, but not necessarily main foodborne source) or pigs (more likely to carry C. coli), and if it were pure plantations.

References

The following references need attention for following: Incomplete, missing country or place of publishing, written unabbreviated. Please follow the journal guidelines when correcting these:

Reference 2, 4, 16, 30, 33, 39, 41.

Author Response

Author comment: The line numbers below refer to the “track changes” version of the manuscript. Our responses begin with “Reply:”

Comments and Suggestions for Authors

Interesting study on the risk associated with open farming systems, investigating the influence of surroundings (agriculture), wild birds, and weather on the spread and likelihood of Campylobacter infection. The study is well-written and commendable.

Reply: Thank you!

Methods

Ln 154: Suggest that ‘labs’ is replaced with ‘laboratories’ in keeping with the scientific format.

Reply: We have replaced “labs” with “laboratories” as suggested (now L148-163).

Ln 161, 166, 176, 177: Please provide the correct format for supplier information, thus city and country

Reply: We have added city and country to the supplier information (now found in L167-178 and L183-187)

Ln 164: Are “physiological” characteristics the correct terminology? Or is it rather the “morphological” characteristics?

Reply: “Morphological” characteristics is correct, and we have modified the text accordingly (now L171-173).

Ln 168: Please insert the ‘bioMérieux’ supplier details for the API kits and complete as per previous comments.

Reply: We have added “bioMérieux” and the supplier information (now L175-178) such that the sentence now reads: “Additionally, for samples collected in 2017 and 2019 that were sent to WADDL, we used an API Campylobacter biochemical test kit (BioMérieux, API Reference Guide, Hazzlewood, MO) to identify Campylobacter isolates to the species level.”

Ln 194-196: Please clarify the statement: “Because… analyses”

Reply: We have clarified the statement at L195-197 (formerly Ln 194-196) such that it now reads: “The percent natural and percent agricultural land cover were strongly inversely correlated (Pearson’s correlation = -0.8, P < 0.0001), so we only used percent agricultural land cover in our analyses.”

Ln 204-205: The survey between sunrise and 1000hr, and the 10-minute period within 1 day collection is not clear. Please rephrase for clarity.

Reply: We have rephrased for clarity such that the sentence now reads (now L216-218): “Each survey was conducted between sunrise and 1000 hr for a duration of 10-mins. Surveys were performed within one day of chicken fecal sample collection.”

Suggest a number below ten to be written out. Please review the manuscript throughout. These appear very inconsistently throughout the manuscript, e.g. ln 205, 280, 296, 431.

Reply: We have screened our manuscript for numbers below ten, and these numbers are now written out.

It is suggested that the author/s use either none, one or two decimals consistently when reporting figures (%). These appear inconsistently throughout the manuscript, e.g. ln 271, 272, 336, 354, 361

Reply: We have screened the document for figures (%) and revised them to be rounded to the tenth decimal place. However, we left the new parameter estimates and confidence intervals in the Results at two decimal places given the scale is low on those numbers.

The reviewer would like to raise a concern with the explanation of the methodology for the landscape statistical models (ln 279-297). The concern is selecting a model that may contribute to a hypothesis and thus be biased to a particular outcome, but looking at the supplementary data that may assist in that regard. However, the table needs more clarity.

Reply: We have revised the legends of all our SI tables and edited our statistical methods for clarity. Our candidate model set did include a null (random effects only model). If the null model is within 2 AICc of the “best” model, then it can be inferred that the covariates tested are not meaningful. However, the null model was 7.1 AICc below the “best” model, suggesting that our parameters tested are meaningful.

At what age was the poultry when samples were collected? Often related to contamination levels.

Reply: We added the average age of poultry sampled to the Methods from L269-276 as follows: “We also estimated the average flock age by consulting with farmers. Most flocks (n =44) contained a cohort of single aged birds, but some flocks (n = 18) consisted of mixed ages. We estimated the average age of mixed-cohort flocks by estimating the percentage of each flock containing each age cohort of birds then weighting each cohort’s age reported to us by the farmers by their representation within the flock. The average age of the broiler flocks was 11.0 wk (±1.1 SE), and layer flocks averaged 84.9 (±7.67 SE) wk.”

We also included age in the PCA since age was confounded with other variables such as production type and rotation frequency. Negative PC1 values are associated with older flocks.

Results

There are very little data related to actual p-values in the text that can support the correlation between agriculture and increased contamination levels of Campylobacter; also the correlation between increased stock levels or otherwise, although Figure 4 describes some confidence levels. The reviewer suggests that this is clarified.

Reply: We have added p-values to the AICc comparison table for readers that would like to see those values. However, it is conventional with AICc to not report p-values, which we’ve adhered to in the text (Arnold 2010; Burnham et al. 2011). We did, however, add parameter estimates and 95% confidence intervals into the main text from the best-supported models that included the respective parameters.

The section is now as follows (L349-359): “Campylobacter spp. prevalence was predicted to increase on farms with higher average wind speeds in the seven days preceding sampling (β = 0.52, 95% CI: 0.21, 0.83; Fig. 4a; Table S3). For every 1 kph increase in wind speed, the odds of detecting Campylobacter spp. increased by 1.2%. Additionally, poultry flocks with higher production type PC1 values (typically younger birds, more rotations per year, higher flock densities, and “broiler” flocks) were predicted to have higher Campylobacter spp. prevalence (β = 0.29, 95% CI: 0.07, 0.51; Fig. 4b). Finally, flocks on farms embedded in landscapes with higher amounts of agriculture were predicted to have higher Campylobacter spp. prevalence (β = 0.52, 95% CI: 0.15, 0.90; Fig. 4c). For every 1.0% increase in % agriculture, the odds of detecting Campylobacter spp. increased by 1.0%. No other variables appeared in models with high support (i.e., ∆AICc < 2.0).”

REFERENCES

Arnold, T.W. 2010. Uninformative parameters and model selection using Akaike’s Information Criterion. Journal of Wildlife Management 74: 1175-1178.

Burnham et al. 2011. AIC model selection and multimodel inference in behavioral ecology: some background, observations, and comparisons. Behavioral Ecology and Sociobiology 65: 23-35.

Ln 359-360: This appears to be a grammatical error.

Reply: We have revised this sentence for grammar and clarity (now L381-384) such that it now reads: “We did not survey large hen houses that used conventional growing methods for this study; however, a meta-analysis performed by Rossler et al. [16] found a higher prevalence of C. jejuni in hens (39.5%) and broilers (32.2%) than we did.”

Discussion

A few thoughts that the reviewer would like to leave with the author/s: It is known that C. jejuni is the most prevalent food-borne bacteria when causing campylobacteriosis in humans. Following this, C. jejuni is also the most prevalent in poultry which makes the former statement more likely. What the authors have shown is that even in the environment, the contamination of poultry with C. coli is unlikely. What would have been very interesting is the sort of agriculture surrounding these farms. If other animals are present in neighbouring farms, either cattle (possible vector for C. jejuni, but not necessarily main foodborne source) or pigs (more likely to carry C. coli), and if it were pure plantations.

Reply: We agree that it would be interesting to examine how the sort of agriculture around the farms impact detection of C. jejuni and C. coli in poultry feces. Unfortunately, we did not collect detailed data on livestock besides the poultry flocks we surveyed. We investigated possible publicly available livestock datasets but did not find anything with high resolution that matches the time frame of our study. For example, the USDA provides county-level livestock densities annually, but counties in our study area are often quite large, so livestock may be far away and largely irrelevant to the focal farm. We also found another dataset called the Gridded Livestock of the World that is 1-km resolution but is based on 2010 (Gilbert 2018). We decided that the resolution of these datasets, mixed with our lacking species identifications for some of our Campylobacter positive samples, would not lead to a sufficiently robust formal analysis. We would be happy to revisit this request if the Reviewers or Editor know of any better datasets we could utilize. 

We have added a statement that future work should investigate this as follows (L405-407): “Future work should examine how neighboring livestock and presence of non-poultry livestock on farms influences prevalence of C. jejuni and C. coli in open-environment poultry, which was beyond the scope of our study.”

REFERENCE

Gilbert, M., Nicolas, G., Cinardi, G., Boeckel, T. P. Van, Vanwambeke, S. O., Wint, G. R. W., & Robinson, T. P. (2018). Global distribution data for cattle, buffaloes, horses, sheep, goats, pigs, chickens and ducks in 2010. Scientific Data 5: 180227.

References

The following references need attention for following: Incomplete, missing country or place of publishing, written unabbreviated. Please follow the journal guidelines when correcting these:

Reference 2, 4, 16, 30, 33, 39, 41.

Reply: We have checked these references and amended them to follow journal guidelines.

Reviewer 2 Report

In the manuscript “Wind speed and landscape context mediate Campylobacter risk among poultry reared in open environments” the authors examine the effect that some parameters have on the presence of Campylobacter spp on poultry reared in open environments.

Please consider these suggestions for improving this manuscript:

Major:

The data presented, support in part, the conclusion of the study. I believe that although the selected parameters were examined in a proper way and in depth, the authors should take into account a wider range of parameters. More specifically:

1.    What was the relationship between the presence of other livestock (mentioned in line 135) and the Campylobacter positive farms?

2.    What kind of animals were farmed in addition to poultry in the selected farms? I think the authors should give some more details on this

3.    What was the biosecurity status of the selected farms (personel, visitors etc)

4.    The authors should give more information with regard to the positive farms. More specifically, there are questions raising such as 

-       Were there any farms that were positive more than one time (if yes, how do you explain that)?

-       Where were all the positive farms situated? (Oregon, Idaho etc)   

Author Response

Author comment: The line numbers below refer to the “track changes” version of the manuscript. Our responses begin with “Reply:”

Comments and Suggestions for Authors

In the manuscript “Wind speed and landscape context mediate Campylobacter risk among poultry reared in open environments” the authors examine the effect that some parameters have on the presence of Campylobacter spp on poultry reared in open environments.

Please consider these suggestions for improving this manuscript:

Major:

The data presented, support in part, the conclusion of the study. I believe that although the selected parameters were examined in a proper way and in depth, the authors should take into account a wider range of parameters. More specifically:

1. What was the relationship between the presence of other livestock (mentioned in line 135) and the Campylobacter positive farms?

Reply: We unfortunately did not keep good records of other livestock present on the farms, but we have added into our Discussion that future work should investigate this (L405-407): “Future work should examine how neighboring livestock and presence of non-poultry livestock on farms influences prevalence of C. jejuni and C. coli in open-environment poultry, which was beyond the scope of our study.”

2. What kind of animals were farmed in addition to poultry in the selected farms? I think the authors should give some more details on this

Reply: As mentioned above, we did not collect detailed information on the number of farmed animals besides poultry on the farms. However, we did have detailed information on a subset of the farms from another project, so we gave examples of non-poultry livestock that were present on the overlapping farms from L139-142: “Nineteen of twenty-seven (70.4%) farms raised other livestock in addition to poultry (e.g., pigs, goats, sheep, cattle, horses, donkeys), 16/27 (59.2%) grew crops in addition to livestock, and 11/27 (40.7%) were USDA certified organic producers.”

3. What was the biosecurity status of the selected farms (personel, visitors etc)

Reply: We have added this information to our Methods section where we describe the farms and give an overview of their biosecurity practices. The new text is as follows (L143-145): “Frequency of biosecurity practices on farms in our study can be found in Fig. S3. Some of the most commonly used practices included fencing, cleaning of housing, and vermin-proof bins.”

We created a new Fig. S3 to show common biosecurity practices on the farms. Farms in this study generally had poor biosecurity practices (e.g., mixed-age flocks, multiple sources of chickens, contact with wildlife).

4. The authors should give more information with regard to the positive farms. More specifically, there are questions raising such as

-       Were there any farms that were positive more than one time (if yes, how do you explain that)?

Reply: We did observe farms to be positive more than once. This may have resulted from repeated introduction of Campylobacter from neighboring farms, livestock within the farm, or from wildlife. Unfortunately, molecular markers would be necessary to determine sources of Campylobacter at that fine a spatio-temporal scale. That was beyond the capabilities of this study. However, we have created a new supplementary figure (Fig. S4) that shows whether Campylobacter spp. was detected in one or more fecal samples for each survey.

-       Where were all the positive farms situated? (Oregon, Idaho etc) 

Reply: We observed Campylobacter positive farms across the entire geographic range of the study (California, Oregon, Washington, and Idaho). The majority of farms (23/27) had Campylobacter present. We now show this in new Fig. S4 and specify this in the Results at L332-334 as follows: “We detected Campylobacter spp. in 250/962 (26.0%) fecal samples screened (Table 1), in 69.4% (43/62) of flocks, and on 85.2% (23/27) of farms. Campylobacter spp. positive farms were located in all states surveyed (Fig. S4).”

Reviewer 3 Report

Thank you for sharing. Interesting idea to incorporate environmental factors into the chain of foodborne pathogene transmission. It fits One Health concept very well.

I have only one small suggestion to indicate at least most numerous species of wild birds that were observed (lines 382-383). This would be of interest for those who carry their studies on wild birds as vectors.

Author Response

Author comment: The line numbers below refer to the “track changes” version of the manuscript. Our responses begin with “Reply:”

Comments and Suggestions for Authors

Thank you for sharing. Interesting idea to incorporate environmental factors into the chain of foodborne pathogene transmission. It fits One Health concept very well.

Reply: We appreciate the interest in our idea.

I have only one small suggestion to indicate at least most numerous species of wild birds that were observed (lines 382-383). This would be of interest for those who carry their studies on wild birds as vectors.

Reply: We have now indicated the three most common wild bird species observed on the farms (see L411-415): “Among 121 species of birds observed on the farms in this study, 42 were recorded interacting with poultry flocks. The most common species were the barn swallow (Hirundo rustica), European starling (Sturnus vulgaris), and house sparrow (Passer domesticus).” A more complete description of wild birds on these farms is the subject of a separate manuscript in preparation.

Round 2

Reviewer 2 Report

I'm satisfied by author's responses and modifications

Author Response

Reviewer 2: I'm satisfied by author's responses and modifications

Author Reply: We thank the reviewer for the helpful suggestions throughout the review process. We appreciate the support of our revision.